## Research Article

Pathways to care; care-seeking behaviors; traditional healers; religious leaders; Uganda; global mental health; psychosis

**Corresponding author:**
Yang Jae Lee;
Email: yangjae.lee@yale.edu

# Pathways to care for psychosis in rural Uganda: Mixed-methods study of individuals with psychosis, family members, and local leaders

Yang Jae Lee[1,2], Kayera Sumaya Nakaziba[2], Sophie Waimon[3], Grace Agwang[4], Kailash Menon[5], Sam Samuel[6], Aaron Damon Dyas[7], Travor Nkolo[8], Haba Ingabire[8], Jason Wykoff[2], Olivia Hobbs[9], Rauben Kazungu[2], Job Basiimwa[2], Robert Rosenheck[1], Scholastic Ashaba[10] and Alexander C. Tsai[10,11,12]

[1]Department of Psychiatry, Yale University, New Haven, CT, USA; [2]Empower Through Health, Iganga, Uganda; [3]School of Public Health Washington University in St. Louis, St. Louis, MO, USA; [4]Uganda Christian University, Mukono, Uganda; [5]College of Arts and Sciences Emory University, Atlanta, GA, USA; [6]Williams College, Williamsburg, MA, USA; [7]The College of the University of Chicago, Chicago, IL, USA; [8]Cavendish University, Kampala, Uganda; [9]College of Letters and Science University of California, Los Angeles, Los Angeles, CA, USA; [10]Department of Psychiatry Mbarara University of Science and Technology, Mbarara, Uganda; [11]Department of Psychiatry Harvard Medical School, Boston, MA, USA and [12]Center for Global Health and Mongan Institute, Massachusetts General Hospital, Boston, MA, USA

## Abstract

**Background:** Low- and middle-income countries (LMICs) bear a disproportionate burden of mental illness, with limited access to biomedical care. This study examined pathways to care for psychosis in rural Uganda, exploring factors influencing treatment choices.
**Methods:** We conducted a mixed-methods study in Buyende District, Uganda, involving 67 in-depth interviews and 4 focus group discussions (data collection continued until thematic saturation was reached) with individuals with psychotic disorders, family members, and local leaders. Structured questionnaires were administered to 41 individuals with psychotic disorders.
**Results:** Three main themes emerged: (1) Positive attitudes towards biomedical providers, (2) Barriers to accessing biomedical care (3) Perceived etiologies of mental illness that influenced care-seeking behaviors. While 81% of participants eventually accessed biomedical care, the median time to first biomedical contact was 52 days, compared to 7 days for any care modality.
**Conclusions:** Despite a preference for biomedical care, structural barriers and diverse illness perceptions led many to seek pluralistic care pathways. Enhancing access to biomedical services and integrating traditional and faith healers could improve mental health outcomes in rural Uganda.

## Impact Statement

This study offers critical insights into the pathways to care for psychosis in rural Uganda, a region where access to biomedical mental health services is severely limited. By examining the experiences and choices of individuals with psychosis, their families, and local leaders, this research uncovers the complex interplay of cultural beliefs, structural barriers, and health-care options that influence treatment decisions in this context. The findings reveal that while there is a strong preference for biomedical care among individuals with psychosis, significant obstacles such as cost, distance, and availability limit access to such services. As a result, many patients pursue pluralistic care pathways, often involving traditional healers and faith-based practitioners. This study not only highlights the challenges faced by people with psychosis in accessing appropriate care but also underscores the importance of integrating traditional and biomedical healthcare systems to improve mental health outcomes. On a broader scale, this research contributes to the global understanding of mental health care in low- and middle-income countries (LMICs). It emphasizes the need for culturally sensitive, accessible, and collaborative healthcare models that respect local traditions while enhancing the availability of biomedical care. The implications of this study extend beyond Uganda, offering valuable lessons for similar settings across sub-Saharan Africa and other LMICs where healthcare infrastructure is limited, and mental health stigma persists. By shedding light on the realities of mental health care in rural Uganda, this research aims to inform policymakers, healthcare providers, and international organizations, encouraging them to develop and implement strategies that bridge the gap between traditional and biomedical care. Ultimately, this work seeks to improve the quality of life for individuals with mental illness in resource-constrained environments, contributing to the global effort to achieve equitable health care for all.

## Introduction

Low and middle-income countries (LMICs) bear a disproportionate burden of mental illness, with approximately 80% of individuals with mental disorders residing in these nations (Rathod et al. 2017). Despite this high prevalence, a substantial number of affected individuals in LMIC do not receive the necessary treatment. For people in LMICs who have severe psychotic disorders, the 12-month biomedical treatment rate is only 29% (compared to 70% in high-income countries), while in many countries in sub-Saharan Africa, it is as low as 13% (World Health Organization 2021). Limited funding, inadequate infrastructure, and a scarcity of trained personnel contribute to the insufficient availability of mental health services in these countries (Molodynski et al. 2017; Rathod et al. 2017; Sarikhani et al. 2021; Kaggwa et al. 2022). In Uganda, there are only 53 registered psychiatrists, most of whom work in urban areas as university lecturers and researchers (Kaggwa et al. 2022). Because of this geographic disparity, people experiencing episodes of psychosis utilize various modalities of care outside of the biomedical health care system to address their mental health challenges (Teuton et al. 2007). Mental illness is also highly stigmatized in Uganda (Rasmussen et al. 2019; Lee et al. 2022a; Lee et al. 2023; Lee et al. 2024).

Health care in Africa has historically been pluralistic, with patients seeking out a range of alternative healers alongside biomedical providers (Beckerleg 1994; Ensink and Robertson 1999; Teuton et al. 2007; Gureje et al. 2015; Tomita et al. 2015). Traditional healers and faith healers are more accessible in rural areas, provide culturally appropriate care, and can offer holistic support (Sorsdahl et al. 2009; Campbell-Hall et al. 2010). Numerous studies have shown that traditional healers and faith healers can accurately identify mental illness symptoms; however, the causes of disorders are attributed to non-medical factors, which influence their treatment approaches (Abbo 2011). Traditional healers associate mental health problems with deviations from cultural norms, ancestral intervention, or even witchcraft (Sorsdahl et al. 2009; Mokgobi 2014; Esan et al. 2019). They employ techniques such as herbal medicine and spiritual guidance to alleviate symptoms (Abbo 2011). In contrast, faith healers, who are primarily denominational or non-denomination Christian clergy, attribute these symptoms to the influence of Satan or sin, relying on prayers and counseling as common interventions (Teuton et al. 2007; Sorsdahl et al. 2009; Mokgobi 2014). In cases of psychosis, which many belief systems in Africa attribute to supernatural causes, these alternative approaches can offer profound meaning to patients and their families (Teuton et al. 2007; Abbo 2011).

Approximately 50% of individuals turn to these alternative healers as their first providers of mental health services (Burns and Tomita 2015). Studies have characterized attitudes of biomedical providers towards traditional and faith healers and vice versa (Gureje et al. 2015; van der Watt et al. 2017; Esan et al. 2019; Gureje et al. 2020), but there has been little research on consumer-level factors that people with psychosis and their families consider when seeking treatment from alternative non-biomedical providers in sub-Saharan Africa. Studies of patients conducted in high-income settings have identified several different motivations for seeking care from alternative non-biomedical providers, including feelings of perceived safety, the ability to exert more control over treatment plans, and having a provider with greater perceived alignment with their own sociocultural beliefs (Astin 1998; Little 2012; Tangkiatkumjai et al. 2020). Few studies have investigated these considerations among patients in LMIC seeking treatment for health

conditions, or among patients in any setting seeking treatment for mental health conditions. Other considerations, such as accessibility and cost of treatments, may influence patients' selection of such providers for the treatment of mental illness in sub-Saharan Africa (James et al. 2018).

Most patients who turn to traditional and faith healers for mental illness report experiences of curative treatment (Abbo 2011). Biomedical researchers and clinicians may view these healing methods as pseudoscientific, violations of human rights, and potentially dangerous (Abbo 2011). In some publicized cases, alternative healers have been documented subjecting patients to unethical behaviors such as shackling and fasting (Opobo 2009; Abbo 2011; Gureje et al. 2020). Although there are known evidence-based psychosocial interventions involving traditional and faith healers, there is little evidence that they change the course of severe mental illness (Gureje et al. 2015; Nortje et al. 2016). Given the pluralistic healthcare beliefs about mental illness in sub-Saharan Africa, understanding the considerations of patients and their families in health-seeking behavior is critical to optimizing mental healthcare overall.

This mixed-methods study examined diverse pathways of care utilized by people with psychosis and their families for the treatment of psychosis and the reasons for their choices. Local leaders were also surveyed given that they often give recommendations on where to go for healthcare. Through a series of structured questionnaires, in-depth interviews, and focus group discussions, we explored the pathways of care utilized by people with psychosis and their families and their attitudes toward different treatment methodologies.

## Methods

### Study site

The study was conducted in the Buyende District of Uganda, a rural district with a population of approximately 450,000 people as of the most recent (2014) census (Uganda Bureau of Statistics 2017). Participants were interviewed from the Irundu, Kagulu, and Bukutula sub-counties(Uganda Bureau of Statistics 2021). Approximately 35–45% of sub-county residents earn under $2 per day; most of the working population is engaged in subsistence farming and animal husbandry (Uganda Bureau of Statistics 2022). The population primarily speaks Lusoga and includes various ethnic groups, with Basoga being predominant.

### Study design and data collection

We conducted 67 in-depth interviews (IDIs) with local leaders ($n = 17$), adults diagnosed with a psychotic disorder ($n = 25$), and family members/caretakers of those with a psychotic disorder ($n = 25$). We also conducted 4 focus group discussions (FGD; consisting of 8 participants each, (total $n = 32$), two of which consisted solely of people with a psychotic disorder. There were 11 individuals who participated in both IDIs and a FGD, so the total sample size was 88. Data collection continued until thematic saturation was achieved, with no new themes emerging from additional interviews. 11 individuals participated in both IDIs and FGDs to provide deeper insights and validate findings across different data collection methods. The combination of IDIs and FGDs was chosen strategically: IDIs allowed for in-depth exploration of personal experiences and sensitive topics, while FGDs facilitated

group dynamics and collective perspectives on community-level issues. To ensure the trustworthiness of the data beyond double coding, we conducted peer briefings among researchers and maintained an audit trail of analytical decisions. All individuals with a psychotic disorder who participated in IDIs and FGDs were administered a structured questionnaire ($n = 41$) assessing places they went for treatment and treatments that they had received.

All participants were approached in the community (typically in clinic waiting rooms, place of residence, or place of work) by research assistants who spoke the local language (Lusoga) and a community health worker who requested their participation in the study. People with a psychotic disorder, including those with schizophrenia, schizoaffective disorder, and bipolar disorder type 1, were selected based on patient registries at health centers or information provided by community health workers. Once identified, people identified with a psychotic disorder were assessed by a Psychiatric Clinical Officer who administered the Mini International Neuropsychiatric Interview (MINI) 7.0.2 to determine if they met the criteria for a psychotic disorder(Sheehan et al. 1998). Those who met criteria were then invited to participate in the study. Family members or caretakers of many of these patients who self-identified as the primary social connection, or one of the primary social connections, of the person with a psychotic disorder were also interviewed. Local leaders were selected from government registries of elected leaders. All participants were 18 years old or older and were read a consent form (in Lusoga) describing the aims of the study and any potential risks and discomforts before obtaining verbal and written consent. Participants who could not write were permitted to indicate consent with a fingerprint in the presence of an impartial witness who was asked to sign the consent form.

During the IDIs and FGDs, we probed overarching topics of mental health treatment/care, stigma, and referral pathways. Some questions varied by participant sub-cohort (local leader, patient, or family) to account for the different experiences of and relationships with people with psychotic disorders. IDIs lasted between 30 and 60 min; FGDs lasted between 30 and 90 min. We audio-recorded IDIs and FGDs, generated transcriptions of the recordings, and translated them into English.

After completing the IDI and/or FGD, the subsample of adults with psychotic disorders (25 participants in the IDIs and 16 participants in the FGDs) were then administered a structured questionnaire. Research assistants entered data from the structured questionnaire directly into the tablet application Kobotoolbox (v2021.2.4, Kobo Inc., Cambridge, MA) on a password-protected and encrypted tablet. Regular backups were performed and stored securely.

### Data analysis

We used the framework method to inductively identify recurring themes in the data(Gale et al. 2013). This method involves systematically coding the data and organizing it into a matrix to facilitate comparison across cases. Each interview transcript was reviewed by two to three study team members, who familiarized themselves with the content and applied a paraphrase, or code, to passages that they interpreted as important. Study team members first coded the first 3 to 5 transcripts of each respondent separately (in groups of 2 or 3). The study team members then met to compare the codes and agreed on codes to apply to all subsequent transcripts. Study team groups then coded the remaining transcripts separately.

Codes were grouped into clearly defined categories, forming the analytical framework. The analytical framework was edited as

additional codes emerged from subsequent transcripts. Finally, the analytical framework was converted into themes and sub-themes. This conversion process involved reviewing the categories in the analytical framework, identifying overarching patterns or concepts that connected multiple categories, and organizing these into broader themes. Sub-themes were then developed to capture more specific aspects or variations within each main theme. This hierarchical organization allowed us to present our findings in a coherent and meaningful structure, highlighting the major insights from our data while preserving the nuances and complexities revealed in our analysis.

## Results

From the IDIs and FGDs, we inductively identified three principal themes: (1) positive attitudes towards biomedical providers, observed among all groups, (2) difficulty accessing biomedical care, and (3) diversity in perceived etiologies of mental illness that led individuals to different types of providers for care. Descriptive data from structured questionnaires helped confirm the qualitative observations.

### Theme 1: Positive attitudes towards biomedical providers

All groups emphasized that they believed biomedical providers were effective in treating mental illness (Supplementary Materials, Appendix A).

> "It (biomedical care) works…. I have seen improvement" – IDI: Man with psychotic disorder, 42 years old

This emphasis was overwhelmingly observed in the data. Families often commented on improvements they witnessed after their family members received biomedical help.

> "As a parent, when you take your patient to hospital, and you see that their condition has improved" – IDI: Family member, 39-year-old woman

Local leaders also praised the effectiveness of biomedical providers, emphasizing not only medications but also counseling.

> "I believe in those who make the medicine so they can also add knowledge to us on how to help us through the challenge. They also give information to people on how to prevent mental illness. And also giving advice" – IDI: Local leader (LC1), 49-year-old man

Positive attitudes towards biomedical care were not exclusionary of positive attitudes towards traditional healers and faith healers. Participants focused their narratives on the psychosocial support that they received from these alternate care providers.

> "The faith healers have tried praying for me so that I can be normal again. They comforted me, prayed for me, and saw to it that my mental state wasn't bad." – FGD: Man with psychotic disorder, 60 years old

However, some patients viewed biomedical care as the only effective treatment.

> "Unless I have taken medicine, there is nothing that can help." – FGD: Man with psychotic disorder, 23 years old

Others endorsed that biomedical care is an important component of care alongside traditional and faith healers.

> "The other role they (traditional healers) have is they collaborate with religious leaders by allowing them to come pray for you at the

hospitals and health facilities when you are admitted." – FGD: Woman with psychotic disorder, 70 years old

Most people with psychotic disorders have sought biomedical care. Most reported it as either the only effective treatment or one of several effective treatments that they had utilized. The survey data are consistent with the predominant themes described in the IDIs and FGDs. Altogether, 21 (51%) participants reported that medication was the most effective treatment, compared with 8 (20%)

who felt that herbs were the most effective and 5 (12%) who felt that prayers were the most effective (Table 1). A total of 19 (46%) participants reported seeking care from a biomedical provider at

**Table 1.** Demographic information

| Sex | |
|---|---|
| F | 30 (73%) |
| M | 11 (27%) |
| **Age** | |
| Mean (SD) | 39.8 (16.4) |
| Median (min, max) | 38 (18, 82) |
| **Marital Status** | |
| Married or cohabiting | 21 (51%) |
| Separated | 5 (12%) |
| Single and never married | 12 (29%) |
| Widowed | 3 (7%) |
| **Religion** | |
| Protestant | 14 (34%) |
| Catholic | 11 (27%) |
| Muslim | 7 (17%) |
| Pentecostal | 5 (12%) |
| Other (specified) | 4 (10%) |
| **Occupation** | |
| Peasant farmer | 25 (61%) |
| Teacher | 4 (10%) |
| Other (type) | 2 (5%) |
| Unemployed | 10 (24%) |
| **Education** | |
| Incomplete primary | 14 (34%) |
| Complete primary | 5 (12%) |
| O-level | 8 (20%) |
| A-level | 2 (5%) |
| Post-secondary | 3 (7%) |
| Vocational training | 2 (5%) |
| No education | 7 (17%) |
| **Diagnosis** | |
| Bipolar disorder | 14 (34%) |
| Psychosis, unspecified | 9 (22%) |
| Schizophrenia | 6 (15%) |
| Schizophreniform disorder | 6 (15%) |
| Schizoaffective disorder | 4 (10%) |
| Brief psychotic disorder | 2 (5%) |

**Table 2.** Descriptive statistics

| Family history of mental illness | |
|---|---|
| Yes | 14 (34%) |
| No | 27 (66%) |
| **Referred to treatment** | |
| Self | 3 (7%) |
| Family member | 23 (63%) |
| Health center | 6 (15%) |
| Religious leader | 2 (5%) |
| **First treatment** | |
| Health center | 19 (46%) |
| Traditional healer | 14 (34%) |
| Religious leader | 8 (20%) |
| **Treatment facilities visited** | |
| One | 6 (15%) |
| Two | 12 (29%) |
| Three or more | 23 (56%) |
| **Received treatment from a health center** | |
| Yes | 33 (81%) |
| No | 8 (19%) |
| **Time before visiting a health center [days]** | |
| Mean (SD) | 1,143.0 (2,644.8) |
| Median (min, max) | 52.0 (1, 12,775) |
| **Time before visiting any treatment [days]** | |
| Mean (SD) | 224.28 (616.2) |
| Median (min, max) | 7.0 (1, 3,650) |
| **Treated with medication** | |
| Yes | 33 (81%) |
| No | 8 (20%) |
| **Current treatment** | |
| Medication | 16 (39%) |
| Herbs | 4 (10%) |
| Prayers | 1 (2%) |
| Combination | 8 (20%) |
| None | 12 (29%) |
| **Most effective treatment** | |
| Medication | 21 (51%) |
| Herbs | 8 (20%) |
| Prayers | 5 (12%) |
| Combination | 1 (2%) |
| None | 3 (7%) |
| Unsure | 3 (7%) |

a health center, while smaller proportions sought care from traditional healers (14 [34%]) and faith healers (8 [20%]) (Table 2).

### Theme 2: Barriers to accessing biomedical care

While biomedical care was highly valued, participants reported many barriers to access (Supplementary Materials, Appendix B). Many relayed concerns about the cost of transportation from their predominantly rural districts to healthcare facilities with psychiatric care.

> "I could not afford transport to get to the health center" – IDI: Man with psychotic disorder, 48 years old

Many participants also voiced concerns about the high cost of biomedical care, specifically the medications that were prescribed.

> "The most effective thing could be buying him more tablets, but we can't, since we can't afford them" – IDI: Family member, 34-year-old woman

The high costs of biomedical care, and the high costs of transportation fees to overcome the geographic inaccessibility, meant that patients and their families had to seek mental health care from geographically more proximal sources (which in the local context tend to be non-biomedical providers like traditional and faith healers). At the same time, however, participants also noted difficulty paying for care from traditional healers and biomedical care.

> "The only role they have is making us poor. Because there is no one who has ever gone to them [traditional healers] and they give him a definite amount. Their things don't end. When you become poor, that is when you will stop going there, but their things don't end." – FGD: Man with psychotic disorder, 63 years old

Overall, access and cost significantly limit patients' abilities to obtain needed mental health care, particularly biomedical care. The study participant narratives were supported by the descriptive data (Table 2). In the sample, 33 (81%) accessed biomedical care and 8 (19%) did not. Among those who ultimately accessed biomedical care, the median time to see a biomedical provider was 52 days (interquartile range [IQR], 7 days–605 days), while the median time to seek any modality of care was only 7 days (IQR, 3 days–240 days).

### Theme 3: Different perceived etiologies of mental illness led individuals to different places for care

The perceived causes of mental illness included witchcraft, genetics, and environmental factors. These influenced participants' choices, and their families' choices, regarding the different types of care arrangements that were sought (Supplementary Materials, Appendix C). Participants who voiced a belief in biological-based causes, such as genetics or illness, indicated that they preferred a biomedical provider:

> "If a family also has a history of madness, it might also look for someone else…. they have been encouraging them to go to hospitals, to meet a psychiatric doctor"– IDI: Local leader (LC 1), 33-year-old man

Participants also endorsed a belief that failure to seek medical care for fever and untreated illnesses could lead to mental health problems.

> "If you get a fever and you fail to go to the hospital, the fever can easily interfere with your reasoning and mental ability" – IDI: Local leader (LC1), 37-year-old man

Additionally, some participants—all of whom had psychotic disorders—perceived drug and alcohol use to be biological causes of mental illness.

> "Another thing people say, it's caused by drugs. For example, marijuana. So they believe when someone takes marijuana, it brings about the issue of mental illness" – IDI: Local leader (LC1), 49-year-old man

For a substantial portion of participants who brought up witchcraft or stress as causes of mental illness, they tended to report traditional or faith healing as being most effective.

> "They made witchcraft which worked upon me… It is the church that has enabled me to survive" – IDI: Person with psychotic disorder, 70 years old female

Most participants – even if they reported beliefs in non-biomedical causes of mental illness (like witchcraft and sin) – endorsed the belief that biomedical providers are most able to diagnose and treat mental illness.

> "They [biomedical providers] have the most effective way of treating mental illness because they first test them and then find out where the problem is or what is the cause of the problem… Health personnel give advice and referrals to patients to go to another health facility or hospital" – IDI: Local leader (LC1), 49-year-old man

Our survey data were consistent with these findings. Most (23 [56%]) participants reported receiving three or more modalities of care. Only 6 (14.6%) of participants reported receiving treatment from a single type of provider. Additionally, for participants who

**Table 3.** Treatments

|  | Traditional healer | Religious leader | Health center |
|---|---|---|---|
| **First treatment:** | | | |
| Counseling | 4 (29%) | 2 (25%) | 5 (26%) |
| Herbs | 14 (100%) | 0 | 0 |
| Ceremonies | 5 (36%) | 0 | 0 |
| Prayers | 2 (14%) | 7 (88%) | 2 (11%) |
| Medication | 3 (21%) | 1 (13%) | 18 (95%) |
| Other | 2 (14%) | 0 | 4 (21%) |
| **Second treatment:** | | | |
| Counseling | 3 (38%) | 6 (55%) | 5 (31%) |
| Herbs | 8 (100%) | 0 | 0 |
| Ceremonies | 5 (63%) | 0 | 0 |
| Prayers | 0 | 11 (100%) | 0 |
| Medication | 0 | 0 | 14 (88%) |
| Other | 1 (13%) | 1 (9%) | 2 (13%) |
| **Third/additional treatments:** | | | |
| Counseling | 0 | 4 (40%) | 6 (19%) |
| Herbs | 6 (100%) | 0 | 0 |
| Ceremonies | 2 (33%) | 0 | 0 |
| Prayers | 0 | 8 (80%) | 1 (3%) |
| Medication | 0 | 0 | 30 (94%) |
| Other | 0 | 1 (10%) | 5 (16%) |

reported receiving care from traditional healers and religious leaders, medications were rarely involved (Table 3).

## Discussion

In this observational mixed-methods study in rural eastern Uganda, we investigated the perspectives of people with psychotic disorders, their family members, and local leaders about treatment for psychosis and pathways of care. We found that most people broadly preferred biomedical care over other types of care but were hampered in their ability to access care due to structural barriers like cost and travel distance. Their perceptions of the etiology of mental illness often led them to seek different modalities of care, suggesting that pluralistic strategies might be useful in improving access and outcomes.

In general, participants expressed broad confidence in the effectiveness of biomedical care for treating mental illness. Few voiced negative opinions of biomedical care. Compared with studies conducted in rural settings in other countries in sub-Saharan Africa, a higher proportion of participants in our study reported that they sought care at a healthcare facility first (rather than seeking care from traditional healers or other practitioners) (Ssebunnya et al. 2009; Burns and Tomita 2015; Kauye et al. 2015; Lilford et al. 2020). The finding that people had difficulty accessing biomedical care was unsurprising, given that there are only 53 registered psychiatrists in Uganda for a population of over 45 million people (Kaggwa et al. 2022). The mixed-methods data suggest that if biomedical care was more readily available and accessible, it would be widely utilized and preferred.

Despite the general preference for biomedical care, participants' perceptions of the etiology of mental illness influenced their pathways to care. Participants attributed mental illness to diverse causes, including biological factors, substance use, stress, and witchcraft. This finding is consistent with existing literature globally and locally, with individuals preferring traditional healers if they believed that the illness was due to spirits and witchcraft and faith healers if they believed that it was due to sin and stress (Audu et al. 2013; Sweetland et al. 2014; Uwakwe and Otakpor 2014; Kisa et al. 2016). Additionally, participants sought multiple modalities of treatment, which is consistent with previous literature on treatment-seeking behavior for mental illness (Burns and Tomita 2015; Badu et al. 2019; Lambert et al. 2020; Pham et al. 2021; Yimer et al. 2023) and for other conditions, such as malaria, HIV, and diabetes (Atwine et al. 2015; Lee et al. 2019;Hooft et al. 2020; Sundararajan et al. 2020; Lee et al. 2022b). Thus, although they may endorse a strong preference for biomedical care, study participants also followed more pluralistic models of care (Beckerleg 1994; Teuton et al. 2007; Sundararajan et al. 2020; van der Zeijst et al. 2023).

The study's strength is its comprehensive approach, combining IDIs, FGDs, and structured questionnaires to capture a nuanced understanding of mental health care seeking behavior. The inclusion of local leaders as key informants adds a valuable perspective on community dynamics and recommendations for healthcare choices. However, the study also has notable limitations. First, it focuses on a specific rural district, where an NGO, Empower Through Health, began providing mental healthcare locally in 2021. Many patients were recruited from the patient registries at Empower Through Health, and this could have artificially inflated the number of people seeking biomedical care thereby explaining the higher proportion of people endorsing seeking

biomedical care compared to other studies. Second, there is a stigma regarding seeing traditional healers for healthcare, as during British colonial rule, officials disparaged traditional customs, and influence in the colonial state depended on conformity to British norms (Mamdani 2006; Hobsbawm 2012). Thus, social desirability bias could have biased participants to endorse more negative views towards traditional healers while endorsing positive views towards biomedical providers. Additionally, researchers were affiliated with Empower Through Health, which could exacerbate social desirability bias, as respondents could have associated the research team with biomedical providers. Third, this study included heterogeneous psychotic disorders, such as bipolar disorder and schizophrenia. There is varying presentation of both illnesses – individuals with bipolar disorder potentially having lengthy periods of stability without medications, which could affect their perspectives and choices heterogeneously. Fourth, the sample size for the quantitative survey data was determined by practical constraints of the study setting and available resources rather than formal power calculations. While our mixed-methods approach allowed for rich qualitative data collection that reached thematic and content saturation, the quantitative survey component may have been underpowered to detect smaller differences in care-seeking behaviors. Lastly, the predominance of female participants (approximately 75% of the sample) warrants discussion. While this gender imbalance may reflect women's greater likelihood to participate in health research and their traditional role as primary caregivers in Ugandan society, it could limit the generalizability of our findings and their applicability to male patients.

These results have important implications for further research and policy in Uganda. It demonstrates that people are willing to utilize biomedical care when available. This finding reinforces the urgency of increasing the accessibility of mental healthcare, as less than 1% of the country's healthcare budget is devoted to mental healthcare (Cooper et al. 2010; Kigozi et al. 2010; Kaggwa et al. 2022). The results suggest that people utilize a pluralistic system of mental healthcare provision, suggesting that a collaborative model could improve outcomes, as demonstrated in clinical trials in Nigeria (Gureje et al. 2020) and Ghana (Ofori-Atta et al. 2018). Future research should focus on the detailed mapping of patient journeys through various care modalities and the evaluation of the integrated care models combining pluralistic strategies. In summary, this study provides a foundation for further research and development of effective pluralistic strategies to address the mental health needs of individuals in low-resource settings of sub-Saharan Africa.

**Open peer review.** To view the open peer review materials for this article, please visit http://doi.org/10.1017/gmh.2024.143.

**Supplementary material.** The supplementary material for this article can be found at http://doi.org/10.1017/gmh.2024.143.

**Data availability statement.** The data that support the findings of this study are available from the corresponding author, YJL, upon reasonable request.

**Acknowledgments.** The team is grateful for the contribution of Buyende District community health workers, who aid in data collection.

**Author contribution.** All authors made substantial contributions to either the conception or design of the work or the acquisition, analysis, or interpretation of data. All authors drafted or revised the manuscript critically for important intellectual content and approved the final version to be published. All authors agree to be accountable for all aspects of the work and ensure that questions related to the accuracy or integrity of any part of the work are appropriately investigated and resolved.

**Financial support.** Empower Through Health provided funding for this research. Dr. Tsai reports receiving funding from the U.S. National Institutes of Health K24DA061696-01.

**Competing interest.** Dr. Tsai reports receiving an honorarium from Elsevier for his work as co-editor-in-chief of the Elsevier-owned journal *SSM – Mental Health*. The other authors have no competing interests to declare.

**Ethics statement.** This research study was approved by the Institutional Review Boards of The AIDS Support Organization, Uganda (TASO-2023-222) and Yale University (2000034605). In accordance with national guidelines, we received approval to conduct the study from the Uganda National Council of Science and Technology (SS1860ES).

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
