## [Reviewer Report]

Thanks for the opportunity to review this manuscript. I believe it can be improved if the authors can address the following:

Abstract

Briefly mention in the abstract if the qualitative sample was determined based on saturation.

The major themes need to stand without being buried by the explanations. Be clear on the three themes e.g. Three main themes emerged: 1) Positive attitudes towards biomedical providers 2) barriers to accessing biomedical care, and 3) Perceived etiologies of mental illness.

Background

Provide a reference for page 5 line 8: “They employ techniques such as herbal medicine and spiritual guidance to alleviate symptoms.”

Be consistent with the in-text references. Some in-text references have full names e.g. page 5 line 26: Jonathan K. Burns and Andrew Tomita 2015

Page 6 line 15. One of the in-text references is missing the date of publication: (Abbo 2011; Kleinman).

Page 6 line 33. Rather say mixed method than qualitative study. If you are presenting only the qualitative bits while the study was a mixed method, you should clearly say it. The abstract says a mixed method while this line is just talking about a qualitative study. This qualitative study examined diverse pathways of care utilized……..

Methods

In the study design and data collection section, also indicate if the sample was based on saturation.

Why did 11 individuals participate in both IDIs and FGDS? What was special about these 11?

Were there some other ways used to ensure the trustworthiness of data apart from the double coding?

Why did you use both IDIs and FGDs? The rationale needs to be presented in your manuscript.

How was the survey sample determined? Does the sample have enough power?

Results

A brief description of descriptive data might be necessary.

Page 10 line 26: Where is appendix A that is written here?

Theme 2 is different from the theme that is in the abstract. The theme in the result section says “Biomedical care is difficult to access” while in the abstract it is written as “barriers to accessing biomedical care.” Be consistent.

---

## [Reviewer Report]

This article provides a clear and concise description of a mixed-methods study analysing pathways to care for individuals in Uganda. The overarching goal of the study is to “shed light” on the realities of mental health care service utilization in Uganda, with a view to eventually designing and implementing strategies that seek to bridge the gap between traditional and biomedical care. I suggest the authors make additions to the descriptive data section, revise the concluding statement/arguments to be more in line with the findings as presented in the body of the study, and include an overview of future research plans.

The introduction/background section does an appropriate job of situating the study within the context of current research on biomedical and traditional practices, citing relevant work from across Sub Saharan Africa (e.g. Gureje) as well as Uganda (e.g. Apopo). No research or reference to current interventions which seek to integrate traditional and biomedical mental care services in Uganda were cited. It is assumed that the authors did not find any, but if they do exist it would be helpful to include references here as part of the context.

The study design and findings are clearly presented and the team does a good job of highlighting potential sources of selection, social preference and other forms of bias. However, neither the site description nor the demographic data provide information about participant ethnicity or linguistic background. Given cultural factors may influence the likelihood of individuals to seek biomedical vs. traditional health services, it seems like this would be helpful information to collect. Is there a reason that such data was not included? In addition, the demographic data appendix indicates that nearly ¾ of the study participants were women, but there is no discussion of this or its potential implications. Why the predominance of female participants? Is this reflective of the context (e.g. is psychosis more prevalent in the female population in Uganda or is it simply that more women are likely to be available to participate in the study)? How might the sex ratio impact the relevance of the study findings for future intervention development?

The conclusion of the study is that “Despite a preference for biomedical care, structural barriers and diverse illness perceptions led many to seek pluralistic care.” The results presented in the article indicate that the majority of service users felt biomedical care was effective and valued, but the study does not appear to ask about preference of one form of care over another. Consequently, I would be cautious of drawing this as the primary conclusion and it may be worth revising the final statement to be more in line with the findings from the data as presented in the body of the paper (e.g. focused on perceived effectiveness of biomedical treatment). It could be that service users feel biomedical treatment is the most effective treatment for reducing psychosis symptoms, but that they nonetheless prefer to access multiple treatment modalities at the same time for other reasons. Alternately, if questions were asked about preference for type of care these should be highlighted to support the concluding statement as its currently written.

Finally, the study reports findings on the percentage of individuals who accessed biomedical care and other care modalities, as well as the amount of time that it took to engage with various types of care. While the study is a good starting point, in future the researchers should consider capturing additional in-depth information on the care seeking journeys of psychosis patients. Specifically, more detailed information on the chronology and pattern of the patient journey and the events which prompt individuals to access one type of service provider vs. another is needed. Seeking health care is rarely an entirely linear process and it’s important to understand if, why and when service users move from traditional to biomedical care, consult both simultaneously on an ongoing basis, or switch to traditional practitioners after accessing biomedical care. Patient journey mapping is one approach that could be used to collect such detailed information that would provide a stronger foundation for the development of pluralistic strategies. While the inclusion of new methods is not feasible at this stage, the article would benefit from a brief description of future research plans that indicate how the team intends to build upon the current findings to build a fuller picture of pathways to care.

Minor Textual Corrections

p.5, line 33 – the study is better described as mixed-methods given the inclusion of quantitative survey data, not solely qualitative

p.17, lines 26-36 – The grammar in this paragraph should be checked to ensure clarity of meaning. The written text alternates between referring to “results” and “findings” in the plural and singular, at times it is unclear if a specific finding is being referenced or simply the study results in general. The last part of line 36 should read “suggesting that a collaborative model could lead to improved outcomes…”

---

## [Reviewer Report]

The authors have adequately addressed all the concerns that I previously had.

I am satisfied with the improved manuscript.

---

## [Reviewer Report]

The authors appear to have taken onboard reviewer comments and made amendments as appropriate. I have not further observations.